# Investigating the impact of enhanced community case management and monthly screening and treatment on the transmissibility of malaria infections in Burkina Faso: study protocol for a cluster-randomised trial

Katharine A Collins,[●][1] Alphonse Ouedraogo,[2] Wamdaogo Moussa Guelbeogo,[2] Shehu S Awandu,[1] Will Stone,[3] Issiaka Soulama,[2] Maurice S Ouattara,[2] Apollinaire Nombre,[2] Amidou Diarra,[2] John Bradley,[4] Prashanth Selvaraj,[5] Jaline Gerardin,[5] Chris Drakeley,[3] Teun Bousema,[1] Alfred Tiono[2]

KAC and AO are joint first authors.
CD, TB and AT are joint senior authors.

For numbered affiliations see end of article.

**Correspondence to**
Teun Bousema;
teun.bousema@radboudumc.nl

## ABSTRACT

**Introduction** A large proportion of malaria-infected individuals in endemic areas do not experience symptoms that prompt treatment-seeking. These asymptomatically infected individuals may retain their infections for many months during which sexual-stage parasites (gametocytes) are produced that may be transmissible to mosquitoes. Reductions in malaria transmission could be achieved by detecting and treating these infections early. This study assesses the impact of enhanced community case management (CCM) and monthly screening and treatment (MSAT) on the prevalence and transmissibility of malaria infections.

**Methods and analysis** This cluster-randomised trial will take place in Sapone, an area of intense, highly seasonal malaria in Burkina Faso. In total, 180 compounds will be randomised to one of three interventions: arm 1 - current standard of care with passively monitored malaria infections; arm 2 - standard of care plus enhanced CCM, comprising active weekly screening for fever, and detection and treatment of infections in fever positive individuals using conventional rapid diagnostic tests (RDTs); or arm 3 - standard of care and enhanced CCM, plus MSAT using RDTs. The study will be conducted over approximately 18 months covering two high-transmission seasons and the intervening dry season. The recruitment strategy aims to ensure that overall transmission and force of infection is not affected so we are able to continuously evaluate the impact of interventions in the context of ongoing intense malaria transmission. The main objectives of the study are to determine the impact of enhanced CCM and MSAT on the prevalence and density of parasitaemia and gametocytaemia and the transmissibility of infections. This will be achieved by molecular detection of infections in all study participants during start and end season cross-sectional surveys and routine sampling of malaria-positive individuals to assess their infectiousness to mosquitoes.

**Ethics and dissemination** The study has been reviewed and approved by the London School of Hygiene and

## Strengths and limitations of this study

► This study evaluates for the first time the impact of enhanced community case management (CCM) and monthly screening and treatment (MSAT) on the transmissibility of malaria from endemic populations to mosquitoes.

► The detailed examination of malaria infections, gametocyte production and infectivity to mosquitoes across two peak transmission seasons and the intervening low transmission season provides uniquely detailed information on the dynamics of parasite carriage and infectivity.

► Intensive monitoring of mosquito exposure in combination with repeated assessments of parasite carriage in the human population allows detailed assessments of the force of malaria infection at different time points during the peak and low transmission seasons.

► A limitation of this current study is that findings from this setting where malaria transmission intensity is high may not translate to areas of lower transmission intensity where the tested approaches may be used in malaria elimination efforts.

► The study specifically aims not to impact transmission or maximise the community effect of the interventions. Rather, the study aims to quantify the impact of enhanced CCM and MSAT on the trajectory of infections experienced by the participating individuals, gametocyte production in these infections and transmissibility to mosquitoes.

Tropical Medicine (LSHTM) (Review number: 14724) and The Centre National de Recherche et de Formation sur le Paludisme institutional review board (IRB) (Deliberation N° 2018/000002/MS/SG/CNRFP/CIB) and Burkina Faso national medical ethics committees (Deliberation N° 2018-01-010).

Findings of the study will be shared with the community via local opinion leaders and community meetings. Results may also be shared through conferences, seminars, reports, theses and peer-reviewed publications; disease occurrence data and study outcomes will be shared with the Ministry of Health. Data will be published in an online digital repository.

**Trial registration number** NCT03705624.

## INTRODUCTION

Malaria continues to be a major public health problem in sub-Saharan Africa. In 2017, there were an estimated 435 000 deaths from malaria globally, 93% in sub-Saharan Africa.[1] Despite this sobering figure and indications that no measurable reduction in global malaria cases has been achieved since 2015,[1] many countries had experienced a considerable decline in malaria transmission prior to this[2 3] meaning malaria elimination and eradication are back on the agenda of the international research community and policy makers.[4 5] To achieve this, interventions that specifically aim to reduce transmission of malaria parasites are high on the priority list of the research agenda for malaria eradication.[6 7] Transmission of malaria begins with the formation of sexual-stage malaria parasites—the gametocytes. In *Plasmodium falciparum* infection, gametocytes develop and mature in the bone marrow and first appear in the circulation 10–12 days after asexual parasites are detected.[8] On release in circulation, male and female gametocytes may persist for several weeks after asexual parasites have been cleared. In chronic infections there is persistent (but fluctuating) production of gametocytes from their asexual progenitors.[9] Once ingested by a blood-feeding female *Anopheles* mosquito, male and female gametocytes activate into gametes that fertilise and ultimately render the mosquito infectious to humans. Gametocytes are present in symptomatic malaria cases and in infections not accompanied by symptoms that are severe enough to elicit treatment-seeking behaviour—so-called 'asymptomatic infections'.[10] Since these asymptomatic infections represent a large proportion of all infections present in malaria-endemic settings,[11 12] asymptomatic parasite carriage may be a major component of the human infectious reservoir for malaria.[13–15] Therefore detecting and treating these infections could be a valuable approach for reducing transmission. It is unclear how many asymptomatic infections start off as symptomatic infections and could potentially be detected and treated by enhanced community case management (CCM). With CCM, access to early diagnosis and treatment is improved by community-based malaria diagnosis. CCM may involve the deployment of malaria posts that improve access to care while still relying on passive detection of infection or, in an enhanced format, may involve active screening for fever.[16 17] In an optimistic scenario, CCM could be used to prevent the majority of infectious days by abrogating infections before they become transmissible to mosquitoes. Alternatively, many incident infections may never elicit symptoms and would therefore remain undetected even during CCM with regular clinical examinations. In this situation, active screening approaches would be needed to identify asymptomatic infections for drug-based targeting to prevent or interrupt the production of infectious gametocytes. Periodic, or monthly screening and treatment (MSAT) approaches aim to detect asymptomatic infections by screening populations regardless of symptoms. Their ability to detect all infections depends on the diagnostic used[13] and in highly endemic settings the impact of MSAT may be transient.[18] The number of infections acquired and the likelihood that infections give rise to clinical symptoms may be influenced by spatial and temporal variations in malaria exposure.[19 20] Thus, detailed entomological assessments may uncover small-scale variations in mosquito exposure that play a key role in determining the detectability and transmissibility of infections in a given area.[21]

In this study, we evaluate the impact of two intensified diagnostic and treatment strategies on the human infectious reservoir of malaria, and compare these to the current standard of care. These diagnostic and treatment strategies aim to (1) Detect and treat symptomatic infections earlier and more comprehensively by enhanced CCM (weekly active screening for fever), and (2) Detect and treat asymptomatic malaria infections by monthly screening with point-of-care rapid diagnostic tests (RDTs) (MSAT).

## METHODS AND ANALYSIS
### Study design

This study is a cluster-randomised trial to determine the impact of two interventions on parasite carriage, gametocyte carriage and infectiousness to mosquitoes. A total of 180 compounds (consisting of three to six enrolled participants) will be randomised to one of three study arms (60 compounds per arm): arm 1 - current standard of care with passively monitored malaria infections; arm 2 - standard of care plus enhanced CCM; or arm 3 - standard of care and enhanced CCM, plus monthly screening and treatment (MSAT), as detailed in box 1.

The study will take place over approximately 18 months covering two high-transmission seasons and the low-transmission season (dry season) in between. Throughout the study, entomological monitoring will take place to evaluate the local mosquito population and potential exposure to infected mosquitoes.

### Gradual implementation of interventions

The aim of the study is to determine the ability of the interventions to detect malaria infections, and the transmission potential of these infections. To maximise coverage and ensure good adherence to intervention protocols the study will be implemented in two phases (figure 1). Phase I will initiate prior to the first transmission season with 30 compounds enrolled per study arm. Phase II will initiate at the end of the first transmission season with an additional 30 compounds enrolled per study arm. Parasitology data and mosquito infection

## Box 1  Study arms

### Arm 1 (control): Standard of care
► Standard of care with passively monitored malaria incidence at local health facilities. Health facilities will be equipped with research staff and adequate supplies to ensure malaria diagnosis and treatment with the first-line antimalarial artemether-lumefantrine (AL) according to national guidelines.
► Seasonal malaria chemoprevention (SMC) in children <5 years of age implemented by the National Malaria Control Programme (monthly treatment with sulfadoxine-pyrimethamine + amodiaquine from July to October).

### Arm 2 (intervention): community case management (CCM)
► Standard of care (as in arm 1).
► SMC (as in arm 1).
► Enhanced CCM for malaria involving weekly active screening for fever using a research-grade thermometer by a trained health worker. A measured temperature ≥37.5°C or reported fever in the last 24 hours will prompt screening with a conventional rapid diagnostic test (RDT). RDT-positive individuals will be treated with AL according to national guidelines.

### Arm 3 (intervention): CCM + monthly screening and treatment (MSAT)
► Standard of care (as in arms 1 and 2).
► SMC (as in arms 1 and 2).
► Enhanced CCM (as in arm 2).
► MSAT regardless of symptoms with a conventional RDT. Screening will be performed by research staff with 25–35 days between screening rounds; RDT-positive individuals will be treated with AL according to national guidelines.

data will be compiled and analysed before the start of the second transmission season (~May 2019), at which time sample size and study procedures may be amended upon ethical approval.

### Study setting
This study will be conducted within the Saponé health and demographic surveillance system (HDSS) area, located 45 km south-west of Ouagadougou, Burkina Faso. This HDSS is part of the INDEPTH Network (a global network of HDSS that provides a more complete picture of the health status of communities in low-income and middle-income countries) and covers a total population of 85 000 living in 10 841 compounds. Compounds are georeferenced and a census of the population is conducted annually. The population is served by 23 local health facilities and a district hospital. This is a region of intense and highly seasonal malaria transmission (predominantly *P. falciparum*) with peak transmission from approximately June to October. Asexual parasite prevalence by microscopy in this population is ~50% in children below 15 years of age[14] and studies in this and other malaria-endemic settings in Burkina Faso indicate that ~80% of all asexual parasite carriers have concurrent gametocytes at highly variable densities.[14 22]

### Enrolment and randomisation
The unit of randomisation in this study is a compound. Per compound a minimum of one and maximum of two

members will be enrolled from each of the three age groups (under 5 years of age, 5–15 years of age and over 15 years of age), so that each compound will provide three to six enrolled participants. All participants must be willing to use the designated local health facilities, and to accommodate indoor and outdoor mosquito sampling for the study duration. Inclusion and exclusion criteria for individual study participants are listed in box 2. From the 2017 census data, an estimated 4700 compounds in the Saponé HDSS contained the required number of participants in each age category and are, in principle, eligible. Up-to-date census data will be used prior to the study to select eligible compounds for enrolment in the Sapone Marche and Pissy local health areas (figure 2). One hundred and eighty compounds will be enrolled and will be divided into 60 groups of three based on proximity. Within each group, one compound will be randomly allocated to each of the three study arms by a computer-generated algorithm. To enable the impact of the interventions to be continually evaluated throughout the study period, the study is designed so that overall transmission intensity is not influenced. To ensure the force of infection remains high ≤50% of all compounds in a given area will be enrolled in the study. This will be achieved by combining each group of three enrolled compounds with three non-enrolled adjacent compounds to form a total of 60 groups (figure 3). Simulations predict that the interventions will have a very small impact on overall transmission in the study area relative to the current transmission intensity (figure 4).

### Procedures
#### Study visits
At the start and end of each season, a cross-sectional survey will be performed concurrently in all enrolled compounds (online supplementary figure S1). These surveys will also include adults that are present in enrolled compounds but not participating in the interventions; additional informed consent will be obtained for this. In these surveys all individuals will have their axillary temperature measured and a finger prick blood sample taken for molecular assessment of gametocyte and parasite carriage (as detailed in table 1). In case of measured fever, the same finger prick blood sample will be used for malaria diagnosis by conventional RDT (first response malaria RDT, specific for *P. falciparum* and/or *P. vivax*, *P. ovalae* and *P. malariae*) and treatment will be offered as per national guidelines. Routine surveys will be conducted throughout the wet and dry seasons to assess parasite carriage, gametocyte carriage and infectivity to mosquitoes. These surveys will be performed on a subset of compounds each week on a rolling basis, so that each compound will be visited every 10 weeks. Finger prick blood samples will be taken and processed as detailed in table 1; one whole blood aliquot will be processed immediately for molecular detection of *P. falciparum* parasitaemia by quantitative PCR (qPCR) targeting varATS.[23] qPCR-positive individuals (ie, those with parasites, not

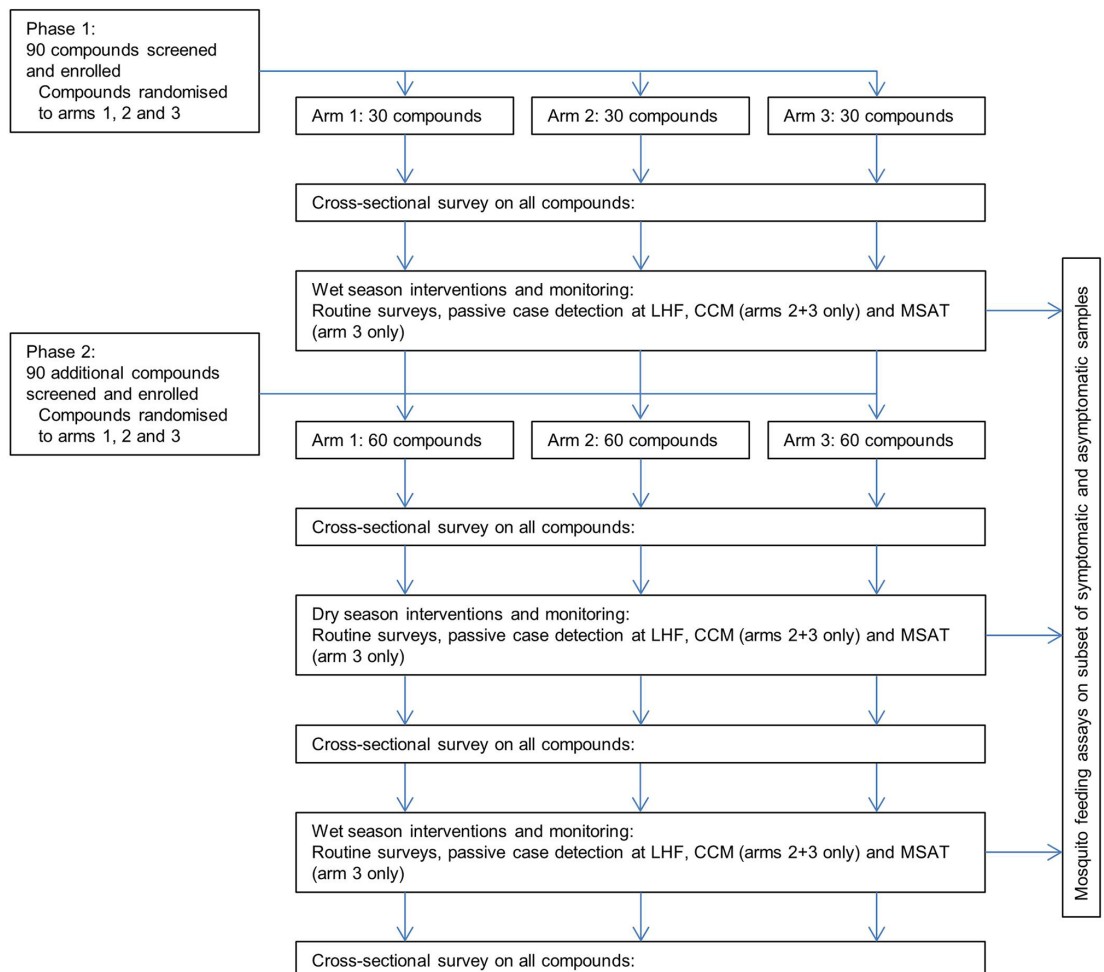

**Figure 1** Study flow diagram. Compounds are enrolled into the study during two phases of recruitment and randomised to the three study arms. CCM, enhanced community case management; LHF, local health facility; MSAT, monthly screening and treatment.

necessarily gametocytes) will be invited to donate blood for mosquito-feeding assays. The maximum number of feeding assays per day will be dependent on logistical feasibility. Asymptomatic infections that are detected by PCR will not be treated if unaccompanied by malaria symptoms, in line with national treatment guidelines. Routine surveys will continue throughout the dry season

to allow a detailed assessment of the number of newly acquired infections when minimal exposure to infected mosquito bites is assumed.

In all three arms, clinical malaria episodes will be passively monitored at the local health facilities. Participants will be identified using study identity cards; for suspected malaria cases, temperature and clinical history will be recorded. Finger prick blood samples will be taken as detailed in table 1. Malaria diagnosis will be performed by RDT and infected subjects will be treated with artemether-lumefantrine (AL). For infections detected by community health workers during enhanced CCM (arms 2 and 3) or MSAT (arm 3) visits, the same procedure is used and samples are collected as detailed in table 1. For RDT or microscopy-positive infections detected at local health facilities or during the CCM or MSAT visits, when logistically feasible, the individuals will be invited to donate a blood sample for mosquito-feeding assays prior to treatment. For all mosquito-feeding assays blood samples will be collected and transported in a portable incubator or thermos flask to the insectary for direct membrane feeding assays (DMFAs).

---

**Box 2    Inclusion and exclusion criteria for enrolled participants**

**Inclusion criteria**
► Participants should be permanent residents of the compound.
► Participants should be willing to participate in repeated assessments of health and infection status and willing to donate a maximum of 37 mL of blood (children <10 years of age) or 52 mL of blood (older individuals) during an 18-month period.

**Exclusion criteria**
► Any (chronic) illness that would affect study participation.
► Pre-existing severe chronic health conditions.
► Current participation in malaria vaccine trials or participation in such trials in the last 2 years.
► History of intolerance to artemether-lumefantrine.

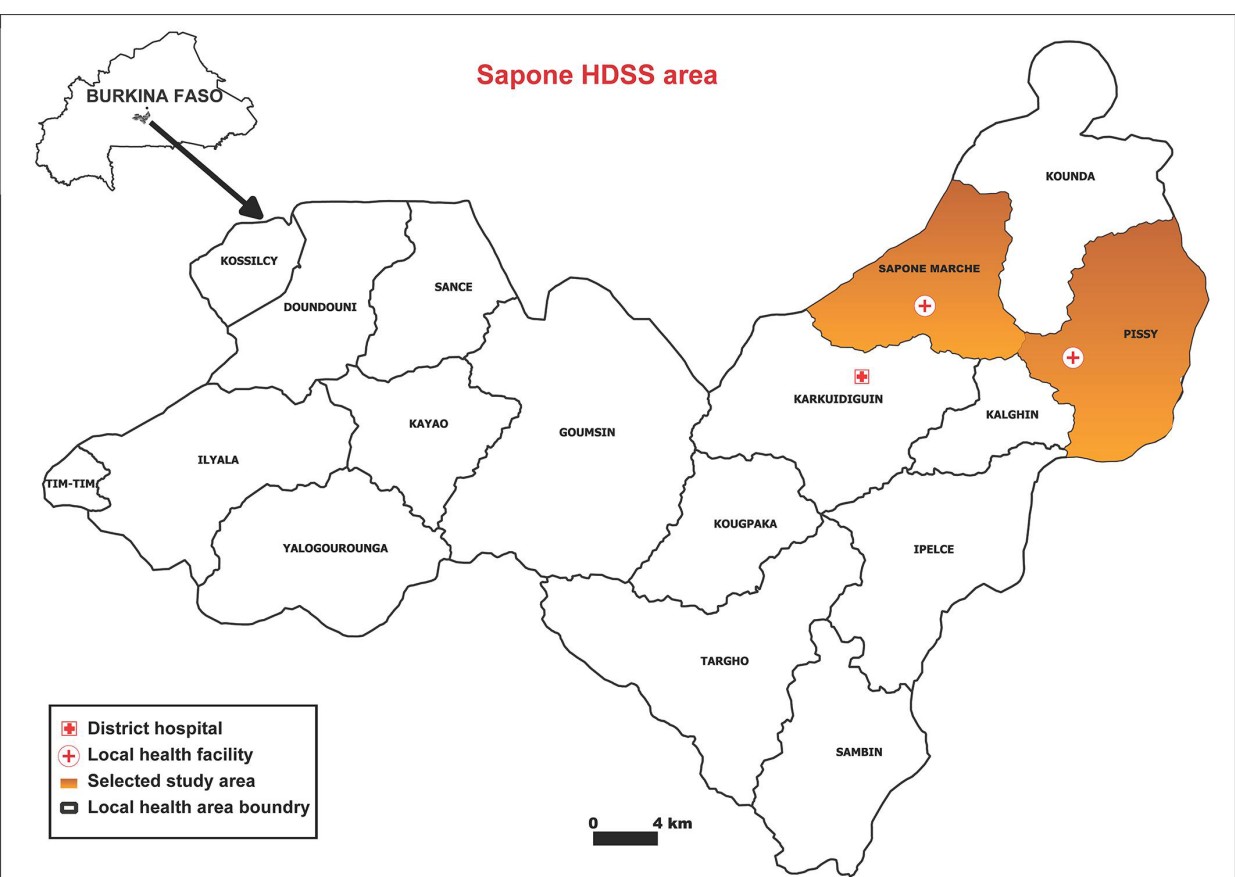

**Figure 2** Map of Sapone health and demographic surveillance system (HDSS) area with the selected study area highlighted in orange.

## Entomology

Detailed assessments of mosquito exposure will be performed at the compound level. Mosquito exposure will primarily be determined using a CDC miniature light trap (Model 512; John W Hock Company, Gainesville, Florida, USA) placed inside one of the sleeping spaces close to the participants sleeping under a net. Sampling will be performed in every compound once every 14 days, throughout the study. To explore outdoor biting and enable more sensitive sampling of sparse mosquito populations, human landing catches will be performed indoors and outdoors from dusk until dawn for a subset of compounds.[24] Due to the low densities of mosquitoes present in the dry season, additional strategies will be used to optimise the potential for mosquito monitoring during that period. Dry season pyrethrum spray catches will be conducted in a selection of non-enrolled compounds to collect indoor resting mosquitoes. These compounds will be sprayed with the insecticide aerosol Kaltox (a mixture of pyrethroids and carbamate containing 0.27% allethrin, 0.2% tetramethrin, 0.17% permethrin and 0.68% propoxur) for 30–45 s; dead and immobilised mosquitoes will be collected after 10 min. In addition, in the absence of water body larval habitats in the dry season an active search of refugia and resting trap collections may be performed close to study compounds. At all collections, factors such as housing type and presence of animals will

be recorded and associations with mosquito exposure will be examined. Throughout the study, presence, size and location of water bodies will be periodically assessed.

Collected mosquitoes will be sorted and counted by sex and species using morphological characteristics, and classified according to blood feeding status and parity (unfed, blood-fed, semigravid and gravid) by microscopy. A subset of mosquitoes will be preserved for further molecular analysis.[25 26] The age structure of the mosquito population will be determined by examination of tracheal distension within the ovaries.[27] Infection status of *Anopheles* caught in CDC traps will be determined using circumsporozoite ELISA[28 29] and or PCR.[30 31]

## Laboratory assessments

DMFAs will be performed to determine infectivity to locally reared female *Anopleles coluzzii* mosquitoes, as described previously.[14] Whole blood collected in lithium heparin tubes will be offered in duplicate to 50–60 starved mosquitoes via an artificial membrane feeder (400–500 µl per feeder). Fully fed mosquitoes will be kept at 27°C–29°C and maintained on a sucrose diet for approximately 1 week (6–8 days). Mosquito infection will be determined by dissection and assessment for the presence and density of oocysts by two independent microscopists. Infected midguts will be stored for molecular confirmation of infection and genotyping for a subset of guts.[14]

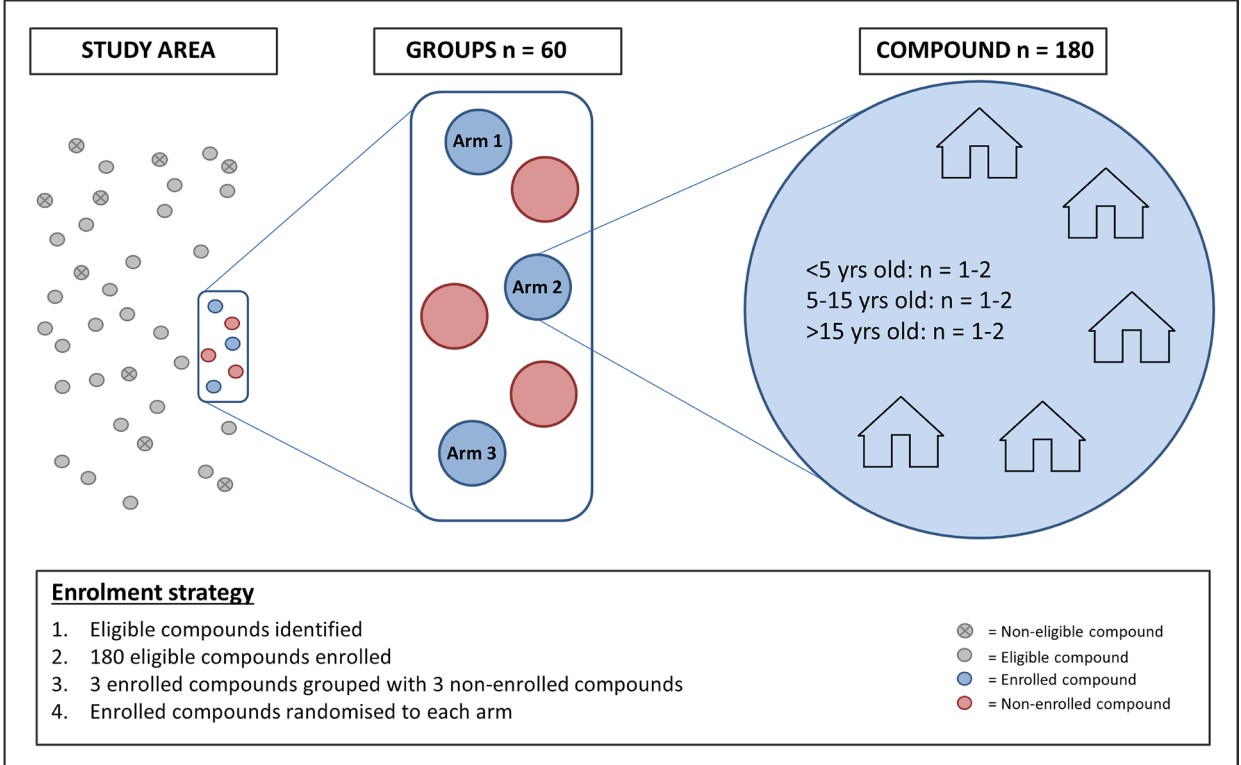

**Figure 3** Study enrolment strategy. Health and demographic surveillance system census data were used to select eligible compounds based on the number and age of occupants. Eligible compounds were enrolled and groups of three enrolled compounds were combined with three non-enrolled compounds, generating 60 groups. Within each group the three enrolled compounds were randomly assigned to each of the three study arms. This strategy is used to ensure (1) An even spread or compounds in each arm over the study area. (2) That ≤50% of all compounds in the study area are enrolled in the study, so overall transmission intensity remains unaffected.

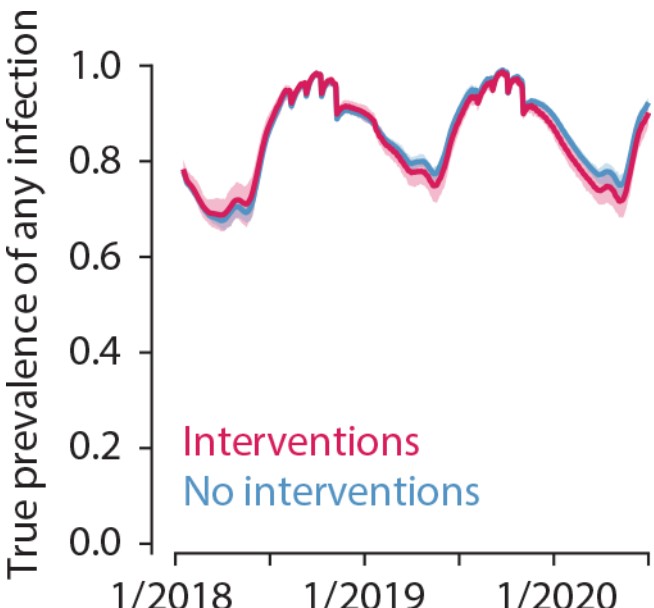

**Figure 4** Interventions have minimal impact on overall transmission in the study area relative to transmission intensity. True prevalence of any malaria infection in simulations with study interventions (red) and without (blue) is similar. Simulated study activities begin in July 2018 and continue through the end of simulation. Lines indicate mean and shaded area the 95% observed interval across 25 stochastic realisations.

Parasite DNA and RNA will be extracted from whole blood samples collected in EDTA tubes. Parasitaemia will be assessed using a qPCR assay targeting the var gene acidic terminal sequence (varATS qPCR)[23] and gametocyte density and sex ratio will be assessed using established PCR protocols.[32] Whole blood samples will be evaluated by highly sensitive (HS)-RDTs and presence and quantification of HRP2 will also be determined using the bead-based Luminex (Austin, Texas, USA) platform.[33] Multiplicity of infection will be determined in a selection of samples using merozoite surface protein (MSP)-genotyping[34] or equivalent techniques.

Red blood cell polymorphisms that are associated with the likelihood of developing blood-stage (clinical) malaria infection, the severity of malaria infection or gametocyte production (haemoglobin C, haemoglobin S, α-thalassaemia and glucose-6-dehydrogenase deficiency)[35] will be measured in baseline samples by multiplex luminex assay.[36] In secondary analyses, antibody responses to parasite and gametocyte antigens will be determined by protein microarray[37] and multiplex bead-based serological assays[38] and associated with parasite carriage, clinical symptoms and infectivity. For a selection of samples functional transmission reducing immunity will be assessed by standard membrane feeding assay using plasma samples collected at the time of field-based DMFA.[39]

**Table 1** Sampling schedule; the table indicates the samples collected at each participant survey/visit

| Sample type/procedure | Cross-sectional survey | Routine survey | Passive case detection | CCM | MSAT | DMFA |
|---|---|---|---|---|---|---|
| Axillary temperature and clinical assessment | X | X | X | X | X | |
| Dried blood spot (DBS) | X | X | X | X* | X | |
| Whole blood in RNA protect | X | X | X | X* | X | X |
| Whole blood for DNA | X | X | X | X* | X | X |
| Rapid diagnostic test (RDT) | X* | | X | X* | X | |
| Highly sensitive RDT (HS-RDT) | | | | | | X |
| Plasma | | | | | | X |
| Blood smear | | | X† | X† | X† | X |

*Sampling only takes place if the subject has fever or history of fever in the last 24 hours.
†Sampling only takes place if the subject has a history of antimalaria treatment in the previous 3 weeks for microscopic confirmation of malarial infection.
CCM, enhanced community case management; DMFA, direct membrane feeding assay; MSAT, monthly screening and treatment.

## Objectives/end points

The overall aim of the study is to determine the impact of the two diagnostic strategies on prevalence and transmissibility of infections. The hypothesis is that the parasite prevalence and density are reduced in arms 2 and 3 compared with the standard of care. More infections will be detected early and may therefore be abrogated before they develop into chronic asymptomatic infections with continuous gametocytaemia. As such the primary end point is parasite prevalence and density in each arm by molecular detection in the end of study cross-sectional survey. Secondary objectives and end points seek to evaluate the impact of the interventions after each season, and to define the contribution and infectivity of gametocytes in these infections, as detailed in box 3. Exploratory objectives aim to understand the detectability and transmissibility of infections, by exploring (1) Relationships between parasitaemia and transmission, and (2) Associations between infectivity and host characteristics, parasite characteristics or mosquito exposure (box 3).

## Data analysis

For primary and secondary end points, parasite and gametocyte prevalences will be compared between arms using multivariate mixed-effects logistic regression models; parasite and gametocyte density will be assessed using multivariate mixed-effects linear regression models on log-transformed values. The number of incident infections will be assessed by Poisson regression and compared between arms. The proportion of infectious participants and proportion of infected mosquitoes will be assessed by logistic regression models. Analyses will take account of correlation of outcomes at the compound level and the matched randomisation scheme. There may be issues with convergence in mixed-effects models given the small number of subjects per compound. In this case, analysis based on compound level summaries will be performed.

For exploratory end points, detected infections will be analysed in relation to the time and duration of infection, parasite and gametocyte prevalence and density, symptoms, complexity of infection, host characteristics, mosquito exposure and infectiousness to mosquitoes. Parasite density will be related to detectability by HS-RDTs. The association between gametocyte density and mosquito infection prevalence and intensity will be determined using an established Bayesian hierarchical model that incorporates the density of male gametocytes and gametocyte sex ratio.[40] In exploratory end points, model fitting will be performed for different strata of infection characteristics (eg, complexity of infection, concurrent asexual parasite density and season) and host characteristics (eg, antigametocyte immune responses, age). Mosquito exposure by indoor and outdoor trapping methods will be associated with the likelihoods of incident infections and opportunities to transmit infections using comprehensive malaria transmission models.[41]

## Sample size calculations and predicted impact of interventions

The hypothesis of the current study is that the parasite prevalence is considerably reduced in arms 2 and 3 compared with the standard of care. Assuming a conservative parasite prevalence of 40% in the control arm, the current sample size should be sufficient to detect a twofold reduction in parasite prevalence with >99% power. This sample size will also allow us to detect the anticipated difference in the proportion of infections that are gametocyte-positive at the moment of detection (secondary objectives 3–5). We assume that ≥80% of all infections detected in cross-sectional surveys (ie, chronic asymptomatic infections) concurrently carry gametocytes.[14] We expect that infections that are detected by CCM or MSAT visits in arms 2 and 3 are of shorter duration and have lower gametocyte prevalence and density.[15]

**Box 3  Objectives and end points**

**Primary objective/end point**

► Parasite prevalence and density between arms by molecular detection at the end of the study cross-sectional survey (time frame: month 18 (end of second transmission season; January–February 2020)).

**Secondary objectives/end points**

► Parasite prevalence and density by molecular detection at the end of year 1 cross-sectional survey (time frame: month 6 (end of first transmission season; January–February 2019)).

► Parasite prevalence and density by molecular detection at the end of the dry season cross-sectional survey (time frame: month 12 (prior to second transmission season; June 2019)).

► Gametocyte prevalence and/or density by molecular methods at the end of study cross-sectional survey (time frame: month 18 (end of second transmission season; January–February 2020)).

► Gametocyte prevalence and/or density by molecular methods at the end of year 1 cross-sectional survey (time frame: month 6 (end of first transmission season; January–February 2019)).

► Gametocyte prevalence and/or density by molecular methods at the end of the dry season cross-sectional survey (time frame: month 12 (prior to second transmission season; June 2019)).

► Gametocyte prevalence and/or density by molecular methods among *Plasmodium falciparum* infections during all visits in the study (time frame: throughout study, an average of 18 months).

► The number of incident infections/clinical incidence detected during community case management involving active weekly fever screening, monthly screening and treatment, and passive case detection (time frame: throughout study, an average of 18 months).

► Infectivity to mosquitoes of *P. falciparum* infections (time frame: throughout study, an average of 18 months).

**Exploratory objectives/end points:**

► The detectability of infections by highly sensitive rapid diagnostic tests (time frame: throughout study, an average of 18 months).

► The relationship between the proportion of infected mosquitoes and gametocyte density (time frame: throughout study, an average of 18 months).

► The impact of infection characteristics on the transmissibility of infections to mosquitoes (time frame: throughout study, an average of 18 months).

► The impact of human host characteristics on the transmissibility of infections to mosquitoes (time frame: throughout study, an average of 18 months).

► Association between total parasite density and gametocyte density (time frame: throughout study, an average of 18 months).

► Malaria transmission potential based on measured gametocyte densities and the association between gametocyte density and mosquito infection rates (time frame: throughout study, an average of 18 months).

► Number of acquired clones based on genotyping (time frame: throughout study, an average of 18 months).

► The duration of infections in the dry season (time frame: months 6–12, between end of season survey January/February 2019 and end of dry season survey June 2019).

► To quantify mosquito exposure in relation to incident infections and transmission opportunities (time frame: throughout study, an average of 18 months).

If we assume ≤50% gametocyte prevalence among these CCM/MSAT-detected infections, monitoring 100 survey-detected and 100 CCM/MSAT-detected infections will result in 99% power to detect this difference (the calculations assumed a coefficient of variation of 0.5 and a significance level of 0.05).

Simulations predict that enhanced CCM will reduce asexual parasite carriage in arm 2 by 70%, 60% and 30% in children under 5 years, 5–15 years and adults >15 years, respectively. The addition of MSAT in arm 3 is predicted to have minimal impact in all age groups (figure 5). Simulations were carried out in Epidemiological MODeling software (EMOD) V.2.15[42 43] an agent-based model of malaria transmission using within-host infection, immunity and infectivity dynamics[41 44] calibrated to field data.[45] The study site census was gridded into 1 km grid cells with a total human population of 7800 individuals and birth and death rates of 39 per 1000 per year, resulting in 17% of the population being under 5, and 25% between 5 years and 15 years. Within a grid cell, the biting rate is surface area dependent but otherwise homogeneous. Humans moved between grid cells at a rate of one trip per person per year, and each trip lasted on average 2 days. Vectors did not move between grid cells. Malaria dynamics are simulated in a highly seasonal setting with an annual entomological inoculation rate (EIR) of 150 in the presence of insecticide treated bed nets (ITNs). Seasonality was based on entomology data collected in 2011. The vector population was modelled as *Anopheles gambiae* mosquitoes with 65% anthropophily and 90% indoor feeding. All grid cells had the same seasonality and transmission intensity. For the simulations, the number of individuals per arm was ~165 during phase I until January 2019 when numbers increased by ~200 individuals per arm for phase II. Means of 500 stochastic realisations are shown. Simulated interventions are as follows:

1. *SMC:* Administration of a curative drug with prophylactic protection for 30 days to children<5 years. Four rounds of SMC carried out each year starting June 30 with 30 days between rounds. Coverage was set at 80% per round and independent between rounds.

2. *ITNs: Estimated age dependent distribution*; 0–5 years=67%, 5–10 years=74%, 10–15 years=68%, 15–20 years=65%, and >20 years=71%.

3. *Passive case detection and health-seeking behaviour:* AL administered to uncomplicated and severe malaria cases with age-dependent distance-based health-seeking. Distances to the two study health facilities in Sapone and Pissy were calculated from the centre of each grid cell with exponential half-distance in coverage of 10 km. Maximum coverage was 50% for children under 5 years, 30% for individuals over 5 years and 80% for severe cases. Patients received treatment within 3 days of onset of clinical symptoms.

4. *Enhanced CCM in arms 2 and 3:* Modelled as continuous weekly active screening where fevers of 37.5°C or higher receive AL, beginning 30 June 2018 with 100% coverage.

5. *MSAT with RDT in arm 3:* Modelled as continuous monthly RDT testing (diagnostic sensitivity 40 asexual

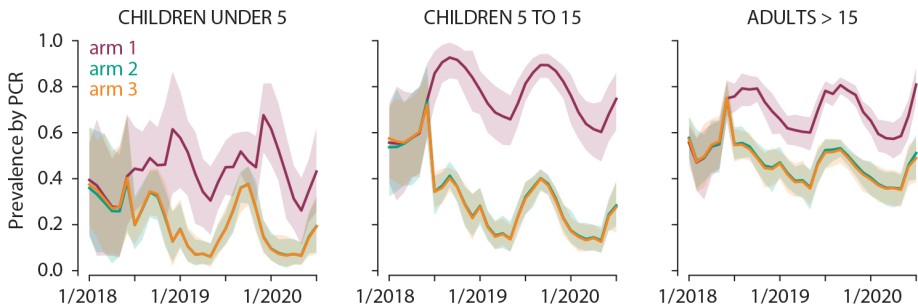

**Figure 5** Simulations predict the impact of interventions in arms 2 and 3 across age groups. Simulation of the impact of enhanced CCM (arm 2, green) and enhanced CCM+MSAT (arm 3, orange) relative to no intervention (arm 1, purple) on parasite carriage in young children, school-age children and adults. Lines indicate mean and shaded area the 95% observed interval of prevalence by PCR with sensitivity 4 parasites/µL across 500 stochastic realisations.

parasites/µL), with RDT-positive individuals receiving AL, beginning 30 June 2018 with 100% coverage.

## Patient and public involvement

Patients and the public were not involved in the design of this study. The findings of the study will be shared with the community via the local opinion leaders. The project is followed by a community meeting to present some of the preliminary findings. Disease occurrence data will be shared periodically with the Ministry of Health and we will work closely with the National Malaria Control Programme. The results may also be shared in conferences, seminars, reports, theses and peer-reviewed publications.

## ETHICS
### Ethical considerations

The purpose of the study and practical consequences of participation will be explained during both community meetings and visits to each household by experienced nurses. Objectives and procedures will be explained verbally and given in printed form in French or a local language. All compound members will be screened by a good clinical practice (GCP)-trained clinician for enrolment criteria, signs of acute disease, and individual informed consent obtained (consent obtained from parent or legal guardian in the case of minors). Malaria infections will not be cleared prior to inclusion in the study but this may be considered later based on results of the first season. Payment will not be given for participation but instead participants will receive financial compensation for travel and time. In addition, each participant enrolled in the study will receive a free long-lasting insecticide-treated bed net and treatment for malaria and other common illnesses free of charge. The blood draw volumes will be minimised: a maximum of 6 mL of blood will be taken at any single time point and the total volume of blood taken during a 6-month period will be lower than 37 mL for individuals enrolled in arm 3, or lower than 22 mL for those in arms 1 and 2. Both the cumulative blood draw volume and maximum blood volume taken in a single sample are within the limits considered as physiological 'minimal risk'.[46] No experimental medication is used in this study; AL is approved in the country as first-line

antimalarial treatment. Study identification numbers will be used on all paper and electronic questionnaires. Some questionnaires will be programmed using OpenDataKit with data entered into password-secured, encrypted tablets. Data will be uploaded daily to an encrypted cloud-based sever. The current protocol version is 2.0 approved November 2018. Any modifications to the protocol which may impact on the conduct of the study, subject safety or potential benefit of the subjects, including changes of study objectives, study design, study procedures, study population or sample size will require a formal amendment to the protocol. Such amendment will be approved by the ethics committees/IRB prior to implementation. The full WHO Trial Registration Data Set information is included in the online supplementary table S1. The trial was submitted to ClinicalTrials.gov(NCT03705624) prior to study initiation on the 19th July 2018. Delays in releasing the registration were related to minor textual changes required to adhere to ClincalTrials.gov formatting requirements and did not change the content of the submission, but resulted in first release online in October.

### Trial oversight, monitoring and quality control

This study will comply with GCP guidelines and be conducted in accordance with the latest South African revision of the Declaration of Helsinki and local regulatory requirements. All study staff will receive a local workshop on GCP before the study starts. Since no experimental treatment is used in the study a data safety and monitoring committee will not be installed; a scientific steering committee will provide technical oversight and will assist with interpretation of data. The London School of Hygiene and Tropical Medicine (LSHTM) is the research sponsor, taking responsibility for the initiation and management of the clinical trial including trial monitoring. The LSHTM Research Governance and Integrity Office has reviewed, assessed and registered the trial and holds Public Liability ('negligent harm') and Clinical Trial ('non-negligent harm') insurance policies which apply to this trial.

Periodically, a subset of compounds will be randomly selected from each arm to be visited by the quality control team to monitor study implementation adherence and unintended effects of study interventions or conduct.

Quality control for mosquito-feeding assays includes regular enrolment of microscopy-positive gametocyte carriers as controls. For this, children aged 10–15 years from non-participating compounds will be screened for the presence of gametocytes by microscopy and up to five gametocyte carriers will be enrolled every 6 weeks. Specific informed consent will be provided for this.

**Author affiliations**
[1]Department or Medical Microbiology, Radboudumc, Nijmegen, The Netherlands
[2]Department of Biomedical Sciences, Centre National de Recherche et de Formation sur le Paludisme, Ouagadougou, Burkina Faso
[3]Department of Immunology and Infection, London School of Hygiene and Tropical Medicine, London, UK
[4]MRC Tropical Epidemiology Group, London School of Hygiene and Tropical Medicine, London, UK
[5]Institute for Disease Modeling, Bellevue, Washington, USA

**Acknowledgements** The authors thank the following individuals for their contributions to the study protocol development: Bronner Gonçalves, Umberto D'Alessandro, Edward Wenger, Lorenz von Seidlein and Tovi Lehman.

**Contributors** AT, TB, CD and KAC developed and designed the study. WMG, AO, SSA, IS, AN, PS, JG, JB, MSO, AD and WS contributed to study design. All authors contributed to protocol development. KAC, AO, CD, TB and AT wrote the manuscript. All authors reviewed and approved the final draft of the manuscript.

**Funding** This study is funded by Bill and Melinda Gates Foundation grant (OPP1173572). KAC and TB are also supported by a fellowship from the European Research Council (ERC-2014-StG639776). AT is supported by an award from the UK Medical Research Council (MRC) and the UK Department for International Development (DFID) under the MRC/DFID Concordat agreement (MR/P02016X/1). The funders, Bill and Melinda Gates Foundation, were involved in study design, but had no role in data collection, analysis, interpretation or reporting.

**Map disclaimer** The depiction of boundaries on the map(s) in this article do not imply the expression of any opinion whatsoever on the part of BMJ (or any member of its group) concerning the legal status of any country, territory, jurisdiction or area or of its authorities. The map(s) are provided without any warranty of any kind, either express or implied.

**Competing interests** None declared.

**Patient consent for publication** Not required.

**Provenance and peer review** Not commissioned; externally peer reviewed.

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
