## [Reviewer comments · BMJ Open]

ARTICLE DETAILS

TITLE (PROVISIONAL)	Study protocol for a cluster-randomized trial investigating the impact of enhanced community case management and monthly screening and treatment on the transmissibility of malaria infections in Burkina Faso
AUTHORS	Collins, Katharine; Ouedraogo, Alphonse; Guelbeogo, Wamdaogo; Awandu, Shehu; Stone, Will; Soulama, Issiaka; Ouattara, Maurice; Nombre, Apollinaire; Diarra, Amidou; Bradley, John; Selvaraj, Prashanth; Gerardin, Jaline; Drakeley, Chris; Bousema, Teun; Tiono, Alfred

VERSION 1 – REVIEW

REVIEWER	Eleanore Sternberg Penn State University, USA
REVIEW RETURNED	22-Apr-2019

GENERAL COMMENTS	I have no major concerns with this manuscript. A few minor comments that might improve clarity:  - A timetable of trial activities might be useful; it would probably allow you to simplify table 4 (objectives and outcomes). But I do recognize that some of that info is captured in figure 1 and, because of the phased design, it might be difficult to show distinct periods for the different activities. - L33, pg 8: citation or explanation for the INDEPTH network? - L26, pg 16: Does everyone in the compound receive an ITN or just the 3-6 people being followed? When/how do they receive that ITN? Is there a plan for monitoring ITN use in the participants? - L46, pg 13: what was coefficient of variation (or intraclass correlation?) and significance level used in the sample size calculations? - SPIRIT checklist 21b: interim analysis (and stopping guidelines) is listed as NA but if I understand L3, pg 8 correctly, there is an interim analysis planned before the start of the second transmission season, which could result in modifications to the protocol.
--

REVIEWER	Michelle Hsiang University of Texas, Southwestern; University of California, San Francisco
REVIEW RETURNED	17-Jun-2019

GENERAL COMMENTS

The authors present the study design for a cluster randomized controlled trial to measure impact of enhanced community case management and monthly screening and treatment of malaria. The goal is to look the impact of the interventions on 1) prevalence and density of infections, and 2) transmissibility of infections. These are important and significant challenges for malaria, but my overall impression of this study is that it doesn't seem to be designed in an optimal way to answer both of these questions. Regarding the first goal, the study does not seem to be optimally designed to maximize impact. Although the study is cluster randomized controlled trial, the unit of randomization is a compound which is quite small and also only a sample of compound members will be enrolled. In order to decrease prevalence at a community level, interventions should be administered to a community, and cover a large portion of the population, or at least a large proportion of the "transmitters" in a population. Also, the authors note that the study is designed in a way to have "minimal impact on overall transmission" as shown in Figure 4. Also, one of the interventions being evaluated (MSAT) has already been shown to not be effective at reducing transmission, so it's unclear why this intervention was selected. With regards to the second goal, to evaluate impact on transmissibility of infections, it's not well described what this means (i.e. how it will be quantified) or why the study was designed the way it was to measure this. I would guess that data from the membrane feeding assays (regarding the infectiousness of different types infection) could be used along with the primary endpoint (parasitemia/density from the cross sectional) to estimate overall transmission intensity. However, transmission studies are only performed during routine surveys and not the final cross-sectional survey (Figure 1). Also, in the methods (page 9), it is written that feeding assays will only be performed with blood from patients that were RDT or slide positive. This seems to imply that the transmissibility/infectiousness of low density infections will not be assessed, yet a low density infections are thought to constitute a major part of the infectious reservoir.

The primary goal of the study is also not clear. In some parts of the paper the primary goal seems to be to measure impact of the interventions (abstract; end of introduction; endpoints section where it is written that the transmissibility goals are 'exploratory'), but then in other parts, it seems the goal is to measure transmissibility (title; first bullet of the "strengths and limitations section"), and then elsewhere it is written that a primary goal is measure detectability of infections (first line of "Gradual implementation of interventions" section).

Other comments (page numbers refer to the page number at the bottom left, not the top left):

Abstract, Page 2 Line 34. The sentence that starts "The recruitment strategy is such that..." is not clear.

Intro, regarding the definition of asymptomatic infections, how about infections that were symptomatic enough to have led to care-seeking, but the level of infection was below the detection threshold of conventional diagnostics?

It would be helpful if the introduction had more background on the current evidence regarding CCM and MSAT, and why these interventions were selected. It would also be helpful to note that

	most asymptomatic infections will be below the detection limit of the conventional malaria tests (used in CCM and MSAT) and explain why a mass drug administration arm was not included. Methods, P. 7 line 27. The unit of randomization and the randomization method should be clearly stated. It seems that the unit of randomization is the compound but then in another part of the paper, there is reference to “clusters” which include compounds that each receive different interventions. The term cluster should be reserved to refer to the unit of randomization. Also, normally in a CRT one would try to minimize contamination. But the study design here almost seems to facilitate contamination. It would be helpful to clarify. Page 8, the first 2 exclusion criteria are not very specific, and it seems that assessment of these 2 criteria could be quite subjective. Regarding AL, are there no other exclusion criteria? e.g. severe malaria or pregnancy in the first trimester? Page 8 31, which RDT will be used? Is it pf specific or does it also detect non pf infections? It would also be good to state in the “study setting” section if this is a Pf predominant setting or other. Page 8. For CCM, will special measures be undertaken to ensure a similar and high quality of implementation by the different community workers across the clusters? Page 8 line 40, It should be noted that the varATS qPCR method is Pf specific. P 13, line 10, is there a reason why poisson models will be used for incident cases, and not negative binomial models? P 14, Does the sample size calculation take into account design effect? Will the analysis take into account differences in baseline factors (i.e. baseline prevalence) and implementation factors (i.e. coverage of study interventions, and receipt of co-interventions (to study participants or other participants in their compound) by the local Ministry or other recent/concurrent studies such as the vaccine trials mentioned in the exclusion criteria) Page 15, how will consent from children and minors be obtained?
--	---

VERSION 1 – AUTHOR RESPONSE

Reviewer: 1

Reviewer Name: Eleanore Sternberg

Institution and Country: Penn State University, USA

I have no major concerns with this manuscript. A few minor comments that might improve clarity:

- A timetable of trial activities might be useful; it would probably allow you to simply table 4 (objectives and outcomes). But I do recognize that some of that info is captured in figure 1 and, because of the phased designed, it might be difficult to show distinct periods for the different activities.

We thank the reviewer for her positive feedback. We have included the suggestion of a timetable of trial activities in the supplementary material.

Figure S1. Timetable of trial activities

CCM = enhanced community case management; MSAT = monthly screening and treatment

- L33, pg 8: citation or explanation for the INDEPTH network?

We have added a description of the INDEPTH network on page 7, line 156.

“This HDSS is part of the INDEPTH Network (a global network of health and demographic surveillance systems that provide a more complete picture of the health status of communities in low and middle income countries) and covers a total population of 85,000 living in 10,841 compounds”

- L26, pg 16: Does everyone in the compound receive an ITN or just the 3-6 people being followed? When/how do they receive that ITN? Is there a plan for monitoring ITN use in the participants?

Each individual enrolled in the study is given an ITN at the start of the study. The use of the ITN is informally monitored at each survey contact when the participants are asked "did they participant sleep under a bed net last night". Though this is not an objective of the study, the information may assist with interpretation of the data. We have clarified this in the manuscript on page 15, line 371.

- L46, pg 13: what was coefficient of variation (or intraclass correlation?) and significance level used in the sample size calculations?

The sample size calculations assumed a coefficient of variation of 0.5 and a significance level of 0.05. This has been added to the manuscript page 13, line 320.

- SPIRIT checklist 21b: interim analysis (and stopping guidelines) is listed as NA but if I understand L3, pg 8 correctly, there is an interim analysis planned before the start of the second transmission season, which could result in modifications to the protocol.

This has been amended to reference this in the new version of the SPIRIT guidelines.

Reviewer: 2

Reviewer Name: Michelle Hsiang

Institution and Country: University of Texas, Southwestern; University of California, San Francisco

The authors present the study design for a cluster randomized controlled trial to measure impact of enhanced community case management and monthly screening and treatment of malaria. The goal is to look the impact of the interventions on 1) prevalence and density of infections, and 2) transmissibility of infections. These are important and significant challenges for malaria, but my overall impression of this study is that it doesn't seem to be designed in an optimal way to answer both of these questions. Regarding the first goal, the study does not seem to be optimally designed to maximize impact. Although the study is cluster randomized controlled trial, the unit of randomization is a compound which is quite small and also only a sample of compound members will be enrolled. In order to decrease prevalence at a community level, interventions should be administered to a community, and cover a large portion of the population, or at least a large proportion of the "transmitters" in a population. Also, the authors note that the study is designed in a way to have "minimal impact on overall transmission" as shown in Figure 4. Also, one of the interventions being evaluated (MSAT) has already been shown to not be effective at reducing transmission, so it's unclear why this intervention was selected.

We appreciate the considerations of the reviewer who has valuable experience in evaluating national elimination programmes and cluster-randomized trials with community endpoints. Our study design is very different in terms of objectives and has a much more academic focus, which is to try and understand the ability of different approaches to detect infections early, and the potential for this to reduce gametocyte carriage and transmissibility. This information will be used to inform future studies that will plausibly have the design the

reviewer seems to propose. The study was designed to address gaps in our current knowledge about the impact of CCM and MSAT on transmissibility of malaria infections which have not been fully examined to date. As is standard practice, the study concept, design and endpoints were reviewed extensively prior to starting. This review was performed by academics and those involved in malaria control programme policy for its scientific rigor and merits to inform policy decisions. Based on these reviews, the design was optimized and, since it is now underway, the design cannot be altered. We have therefore not made changes to study design in the current manuscript. To ensure that the aim of the study is clear and avoid confusion, the protocol states in several places that the aim is not to have any impact on transmission. For example, the abstract now reads –

“The recruitment strategy aims to ensure that overall transmission and force of infection is not affected so we are able to continuously evaluate the impact of interventions in the context of on-going intense malaria transmission”.

For further clarity we have also added a bullet point to the strengths and limitation of the study, indicating that -

“The study specifically aims not to impact transmission or maximize the community effect of the interventions. Rather, the study aims to quantify the impact of enhanced community case management and monthly screening and treatment on the trajectory of infections experienced by the participating individuals, the gametocyte production in these infections, and transmissibility of these infections to mosquitoes”.

With these changes, we are confident that the reviewers concern about the study not being optimally designed to maximize impact are now clarified.

With regards to the second goal, to evaluate impact on transmissibility of infections, it's not well described what this means (i.e. how it will be quantified) or why the study was designed the way it was to measure this. I would guess that data from the membrane feeding assays (regarding the infectiousness of different types infection) could be used along with the primary endpoint (parasitemia/density from the cross sectional) to estimate overall transmission intensity. However, transmission studies are only performed during routine surveys and not the final cross-sectional survey (Figure 1). Also, in the methods (page 9), it is written that feeding assays will only be performed with blood from patients that were RDT or slide positive. This seems to imply that the transmissibility/infectiousness of low density infections will not be assessed, yet a low density infections are thought to constitute a major part of the infectious reservoir.

We agree with the reviewer that low-density infections are potentially important for transmission. As such an aim of the study is to perform mosquito feeding experiments based on qPCR with an approximate sensitivity of 0.1 parasites/uL, well below the threshold for detection by RDT or microscopy. This is clarified on page 8 line 200:

“qPCR positive individuals (i.e. those with parasites, not necessarily gametocytes) will be invited to donate blood for mosquito feeding assays. The maximum number of feeding assays per day will be dependent on logistical feasibility”

Evaluating the transmissibility of infections in the study population is defined as a secondary objective. Mosquito feeding assays are logistically challenging and require a large volume venous blood, limiting the sampling opportunities, thus we aim to perform as many feeding assays as logistically feasible. Feeding assays will be performed on malaria infected subjects from a number of surveys, either qPCR positives identified during routine surveys, or RDT positives identified during either MSAT surveys, CCM surveys, or passive case detection at the local health facilities. This is clarified on page 9 line 212:

“For RDT or microscopy positive infections detected at local health facilities or during the CCM or MSAT visits, when logistically feasible the individuals will be invited to donate a blood sample for mosquito feeding assays prior to treatment.”

The cross-sectional surveys are too large in terms of sampled individuals to perform mosquito feeding assays. However, we will be able to estimate transmissibility in these surveys by the unsurpassed number of mosquito feeding assays performed in this study in the above-mentioned surveys that will relate gametocyte density to mosquito infection prevalence and allow inferring transmissibility from gametocyte density estimates measured in cross-sectional surveys - defined as exploratory objective as detailed on page 12 in Table 4, exploratory objectives/endpoints 2 and 6, and also on page 13 line 301.

The primary goal of the study is also not clear. In some parts of the paper the primary goal seems to be to measure impact of the interventions (abstract; end of introduction; endpoints section where it is written that the transmissibility goals are ‘exploratory’), but then in other parts, it seems the goal is to measure transmissibility (title; first bullet of the “strengths and limitations section”), and then elsewhere it is written that a primary goal is measure detectability of infections (first line of “Gradual implementation of interventions” section).

We thank the reviewer for highlighting this. The overall aim of the study is to assess the impact of the interventions on the “prevalence and transmissibility of infections” and when combined this is also referred to as the “malaria infectious reservoir”. This aim is assessed via a number of specific objectives/endpoints detailed in Table 4 on page 12. To clarify this we have amended the manuscript on page 11, line 273, and we have searched the manuscript for other misleading references to alternative aims and objectives/endpoints, but we cannot find further discrepancies.

Page 11, line 273: “The main study objective is to determine the impact of the two diagnostic strategies on prevalence and transmissibility of infections. The hypothesis is that the parasite prevalence and density are reduced in arms 2 and 3 compared to the standard of care. More infections will be detected early and may therefore be abrogated before they develop into chronic asymptomatic infections with continuous gametocytemia. As such the primary endpoint is parasite prevalence and density in each arm by molecular detection in the end of study cross-sectional survey. Secondary objectives and endpoints seek to evaluate the impact of the interventions after each season, and to define the contribution and infectivity of gametocytes in these infections, as detailed in Table 4. Exploratory objectives aim to understand the detectability and transmissibility of infections, by exploring i) relationships between parasitemia and transmission, and ii) associations between infectivity and host characteristics, parasite characteristics, or mosquito exposure (Table 4).”

For the reviewer, we hope it is helpful that we summarise and explain the aims here in different wordings - The overall aim of the study is to assess the impact of the interventions on the “prevalence and transmissibility of infections”. The primary objective is to assess the impact of the interventions on the parasite prevalence and density. This is the broadest measure of the “transmissibility of infections” ie. If the parasite density is below a certain level it will not contribute to transmission and thus we can assess the proportion of individuals with parasite densities above a certain threshold. The secondary and exploratory objectives address this overall aim “transmissibility of infections” in more detailed and specific ways. If the reviewer sees further inconsistencies are we will be happy to alter and clarify.

Other comments (page numbers refer to the page number at the bottom left, not the top left):

Abstract, Page 2 Line 34. The sentence that starts “The recruitment strategy is such that...” is not clear.

In response to the reviewers comment, we have amended this sentence for clarity on page 2, line 46.

“The recruitment strategy aims to ensure that overall transmission and force of infection is not affected so we are able to continuously evaluate the impact of interventions in the context of on-going intense malaria transmission”.

Intro, regarding the definition of asymptomatic infections, how about infections that were symptomatic enough to have led to care-seeking, but the level of infection was below the detection threshold of conventional diagnostics?

This is an interesting point and relates to the likelihood that, in our study area of intense malaria transmission, parasite densities below the threshold density that allows detection by RDT would elicit malaria symptoms. Asymptomatic infections refer to infections that do not elicit treatment seeking and thus normally remain undetected with conventional case management approaches. Symptomatic infections are those that cause symptoms with detectable infections by routine diagnostics. This is conventional practice and we are not aware of any literature suggesting that in areas of intense malaria transmission, submicroscopic infections are a relevant cause of clinical disease.

It would be helpful if the introduction had more background on the current evidence regarding CCM and MSAT, and why these interventions were selected. It would also be helpful to note that most asymptomatic infections will be below the detection limit of the conventional malaria tests (used in CCM and MSAT) and explain why a mass drug administration arm was not included.

In response to the reviewers comment, we have expanded the introduction to include more background on CCM and MSAT. We specifically mention the dependence of MSAT on sensitive diagnostics to detect all transmissible infections. Whilst MSAT has the overarching aim of reducing population parasite prevalence, to our knowledge, no studies have been conducted directly evaluating the impact of CCM and MSAT on the “transmissibility of infections” in this manner, hence the reason for performing this study. Both CCM and MSAT represent potential approaches that the malaria control programme of Burkina Faso could implement if proven effective. Therefore, the aims and interventions were guided by the need for more data on the impact of “active enhanced community case management (CCM)” and “Monthly screening and treatment” in this context. We understand the reviewers point about MDA, but as indicated above, the aim of this study is not to influence overall transmission. Actually, the specific aim is NOT to influence transmission and examine the detectability of infections in the context of intense transmission. This has been clarified in the strengths and limitations of the study sections. Because we did not aim to influence overall transmission, MDA was not included here. However, as part of a second study in an area of much lower transmission intensity in The Gambia, an MDA-arm is included and a specific aim is to reduce overall malaria transmission. As clarified above, the aim of the current study is different.

Methods, P. 7 line 27. The unit of randomization and the randomization method should be clearly stated. It seems that the unit of randomization is the compound but then in another part of the paper, there is reference to “clusters” which include compounds that each receive different interventions. The term cluster should be reserved to refer to the unit of randomization. Also, normally in a CRT one would try to minimize contamination. But the study design here almost seems to facilitate contamination. It would be helpful to clarify.

In response to the reviewers comment, we have clarified this. The study is not designed as a conventional cluster randomised trial that aims to compare cluster-level differences in outcomes. Our specific design is to ensure that the impact of the interventions can be continuously evaluated in the context of on-going intense transmission. The unit of randomisation is stated clearly as the “compound” on page 7, line 166 and the randomisation method is stated on page 7, line 176. The compounds are divided into groups of 3 based on location and proximity. This is essential to facilitate homogeneity between each study arm to account for geographical and mosquito exposure variations over the study area. The authors cannot find reference to the cluster as the unit of randomisation in the text, but did find the word cluster in Figure 3. This has now been changed to “group”.

Page 8, the first 2 exclusion criteria are not very specific, and it seems that assessment of these 2 criteria could be quite subjective. Regarding AL, are there no other exclusion criteria? e.g. severe malaria or pregnancy in the first trimester?

The approach we have adopted is standard and in line with all clinical trials performed at the CNRFP study sites in Burkina Faso, and has also been approved by the regulatory and ethical bodies at LSHTM and local and national ethics committees in Burkina Faso. The first two exclusion criteria are indeed not very specific as mentioned by the reviewer, and this was intentional to enable the clinician enrolling the participants to exercise their judgment since it would be extremely difficult to list all the potential illnesses/conditions that could prevent the subject from participating in the study, many of which would be incredibly rare. The study is underway and thus it would not be possible to amend the exclusion criteria at this stage

With regards to the AL treatment, this has also been considered. Severe malaria and pregnancy first trimester are “transient/temporary” conditions which do not preclude the subject from being enrolled in or from participating in the study, but of course at the time of treatment, the clinician would determine whether it is safe to administer AL and treatment would proceed in accordance with the standard local treatment practice. It should be noted that in Burkina Faso, severe malaria is not an absolute contraindication for the use of AL which is prescribed as relay to the intravenous infusion as soon as the oral route is possible.

Page 8 31, which RDT will be used? Is it pf specific or does it also detect non pf infections? It would also be good to state in the “study setting” section if this is a Pf predominant setting or other.

We have clarified this and have added the RDT type to page 8 line 193, and the predominant parasite in the study area and page 7, line 161.

Page 8. For CCM, will special measures be undertaken to ensure a similar and high quality of implementation by the different community workers across the clusters?

This is a valuable point. During CCM symptoms/participant information are recorded and auxiliary temperature is taken. All community health workers received full training before the start of the study. All study visits are recorded electronically and reviewed by a senior community health worker and study physician daily and periodic QC of field activities takes place to ensure procedures are being adhered to as mentioned on Page 16, line 405.

Page 8 line 40, It should be noted that the varATS qPCR method is Pf specific.

We have amended the sentence on page 8 in line 199 to specify this.

P 13, line 10, is there a reason why poisson models will be used for incident cases, and not negative binomial models?

Poisson models will be used in this study for incident rates as this is the standard approach for this type of analysis and has been employed for many of our previous studies. Negative binomial models are used for counts rather than rates.

P 14, Does the sample size calculation take into account design effect? Will the analysis take into account differences in baseline factors (i.e. baseline prevalence) and implementation factors (i.e. coverage of study interventions, and receipt of co-interventions (to study participants or other participants in their compound) by the local Ministry or other recent/concurrent studies such as the vaccine trials mentioned in the exclusion criteria)

These are valid points. The model that was used to simulate the impact of the intervention on overall transmission intensity will be used for the analyses the reviewer proposes and incorporates all baseline data. All other analyses will also be adjusted for relevant covariates and account for hierarchical structures in data collection and design effect. This has been clarified in the revised manuscript on page 13, line 294.

“Analyses will take account of correlation of outcomes at the compound level and the matched randomisation scheme”

Page 15, how will consent from children and minors be obtained?

Consent for minors will be provided by a parent/legal guardian. We have amended the sentence on page 15, line 368 to specify this.

“consent obtained from parent or legal guardian in the case of minors”